# Feline Coronavirus in Northern Vietnam: Genetic Detection and Characterization Reveal Predominance of Type I Viruses

**DOI:** 10.3390/v17020188

**Published:** 2025-01-28

**Authors:** Hieu Van Dong, Witsanu Rapichai, Amonpun Rattanasrisomporn, Jatuporn Rattanasrisomporn

**Affiliations:** 1Center for Advanced Studies for Agriculture and Food, Kasetsart University Institute for Advanced Studies, Kasetsart University, Bangkok 10900, Thailand; dvhieuvet@vnua.edu.vn; 2Faculty of Veterinary Medicine, Vietnam National University of Agriculture, Trau Quy Town, Gia Lam District, Hanoi 131000, Vietnam; 3Department of Companion Animal Clinical Sciences, Faculty of Veterinary Medicine, Kasetsart University, Bangkok 10900, Thailand; 4Interdisciplinary of Genetic Engineering and Bioinformatics, Graduate School, Kasetsart University, Bangkok 10900, Thailand; fgraapr@ku.ac.th

**Keywords:** cat, feline coronavirus, genetic detection, PCR, Vietnam

## Abstract

The objectives of this study were to investigate feline coronavirus (FCoV) infection in domestic cats raised in northern Vietnam and to conduct genetic characterization of the FCoV strains circulating across Vietnam. A total of 166 samples were collected from sick and healthy cats in four cities and provinces in northern Vietnam. The FCoV genome was examined using PCR. Sequencing of the partial spike gene was performed. In total, 19 (11.45%) out of 166 samples were positive for the FCoV genome. The genetic analysis of the partial spike gene region indicated that the nucleotide identity of the nine FCoV strains obtained in this study ranged from 85.5% to 99.16% and belonged to type I. No mutations of nucleotides were found at sites 23,531 and 23,537 in the S gene sequences. The furin cleavage site of the nine Vietnamese FCoV strains had the R/G-R-S/A-R-R-S motif.

## 1. Introduction

Feline coronavirus (FCoV), referred to as Alphacoronavirus 1, is classified within the Alphacoronavirus genus, under the order Nidovirales, subfamily Coronavirinae, and family Coronaviridae [1]. The virion structure consists of two fundamental components, the nucleocapsid and the outer envelope, which collectively function to provide stability and preserve the RNA viral genome [2]. Spherical FCoV virion morphology typically exhibits a moderate degree of pleomorphism, with diameters ranging from 80 to 120 nm. Significantly, these viral particles exhibit conspicuous club-shaped surface projections or spikes, with dimensions approximately ranging from 12 to 24 nm. These structural elements confer upon the virus its hallmark coronal or crown-like morphology, giving it the designation coronavirus [3].

FCoV can be categorized into two types I and II, based on their characteristics when grown in laboratory conditions, their genomic properties, and their reactions to antibodies [4]. The behavior and features of these two types differ significantly, especially in terms of how they interact with receptors and adapt to cell cultures [5]. Type I FCoV is considered the more ancestral form, while type II FCoV results from a genetic recombination event between type I FCoV and canine coronavirus (CCoV) [6]. Although type II FCoV is less commonly found in natural surroundings, it is relatively easy to isolate and cultivate in vitro cell cultures [7]. This recombination often includes genetic segments like the spike protein from CCoV, along with varying amounts of adjacent 3a, 3b, and 3c genes, as well as envelope genes [8,9]. Both type I and type II FCoV can exist in less-virulent forms as well as in forms associated with Feline Infectious Peritonitis (FIP). Type I FCoV is more prevalent in most regions worldwide, with reported prevalence rates ranging from 80% to 95% [7,10]. Additionally, FCoV can be categorized into two biotypes: feline enteric coronavirus (FECV) and feline infectious peritonitis virus (FIPV). FECV is known for its high level of contagiousness [8]. When FECV causes clinical manifestations, they typically manifest as mild enteritis. In contrast, FIP, first documented in the scientific literature in 1966, is attributed to FIPV, a highly lethal systemic disease that affects cats globally [9]. These genetic changes can result in a shift from mild enteritis to a severe, symptomatic disease with fatal consequences.

FCoV consists of an outer envelope and a single-stranded RNA genome with a positive-sense orientation. The FCoV genome is approximately 30 kb in length and comprises 11 open reading frames (ORFs). These ORFs code for various proteins, including a polyprotein with RNA synthesis function (1a, 1b), four structural proteins (spike protein (S), envelope protein (E), membrane protein (M), and nucleocapsid protein (N)), and several non-structural accessory proteins (3a, 3b, 3c, 7a, 7b). The untranslated region (UTR) at the 3′ end of the FCoV genome is highly conserved [11]. Previous research has highlighted that mutations occurring at positions 23,531 and 23,537 bp within the S gene may increase the likelihood of FIP development [12]. In contrast, some researchers have suggested that mutations at these two sites can alter the virus’s tissue preference but may not necessarily be linked to FIP [13]. Nevertheless, the precise mechanism responsible for the transition from FECV to FIPV remains poorly understood to this day [14].

In Vietnam, few studies on FCoV infection have been reported. This study was conducted to investigate FCoV infection among Vietnamese domestic cats in northern Vietnam and further analyze genetic characterization based on the partial spike (S) gene sequences of viral strains obtained. The current data generated herein hold the potential to offer valuable insights for the diagnosis, prevention, and control strategies pertaining to FCoV infection across Vietnam.

## 2. Materials and Methods

### 2.1. Ethics Statement

This study was approved by the Committee on Animal Research and Ethics of Vietnam National University of Agriculture (CARE-2022/08). All procedures involving animals were performed after obtaining ethics approval from the Committee on Animal Research and Ethics of Vietnam National University of Agriculture. Informed consent was acquired from the cat owners before sample collection and data release.

### 2.2. Samples

In this study, fecal samples were collected from 166 healthy and sick cats aged between 3 and 36 months visiting veterinary clinics for various purposes such as spa services, treatment, and vaccination in four locations: Hanoi (*n* = 64), Bacgiang (*n* = 35), Hungyen (*n* = 48), and Hanam (*n* = 19). The fecal samples were collected using the swab method and placed in sterile tubes. The samples were then stored in PBS at 2 to 8 °C and sent to the Vietnam National University of Agriculture. Next, the fecal samples were homogenized in phosphate-buffered saline and stored at −80 °C until use.

### 2.3. Total RNA Extraction and cDNA Synthesis

Total RNA was extracted from homogenized samples using the Viral Gene-spin DNA/RNA Extraction kit (Intron, Korea), following the manufacturer’s instructions. Total RNA was preserved at −30 °C until use. The cDNA was synthesized from RNA using M-MLV reverse transcriptase (Promega) and random primers. PCR was performed at 37 °C for 1 h and 72 °C for 5 min. The cDNA product was then stored at −30 °C until use.

### 2.4. PCR, Nested PCR, and Nucleotide Sequencing

PCR was performed to amplify a target gene using specific primers (Table 1) and GoTaq^®^ Green Master Mix (Promega, Madison, WI, USA). A volume of 25 µL PCR reagent consisted of 12.5 µL of GoTaq^®^ Green Master Mix, 1 µL of each forward and reverse primer (10 µM), 8.5 µL of distilled water, and 2 µL of cDNA. The PCR process for FCoV detection was performed at 90 °C for 5 min, followed by 40 cycles at 94 °C for 50 s, 55 °C for 1 min, and 72 °C for 1 min, with a final extension at 72 °C for 10 min. The PCR product of 223 bp was electrophoresed on a 1.2% agarose gel and visualized under ultraviolet light.

To investigate the typing of FCoV strains obtained in this study, nested PCR was used. Three (Iffs, Icfs, and Iubs) and three (nIffles, nIcfs, and nIubs) primers (Table 1) were used for the first and second rounds of nested PCR, as previously described [15].

Nested PCR was used to amplify a portion of the S gene with a size of 142 bp using two sets of primers, FCoV-Mut-F5/R5 and FCoV-Mut-F6/R6, and 156 bp using primers FCoV-Mut-F7/R7 and FCoV-Mut-F8/R8 (Table 1). PCR components were similar to those described above. Nested PCR reactions for primers FCoV-Mut-F5/R5 and FCoV-Mut-F6/R6 were performed at 94 °C for 5 min, followed by 30 cycles at 94 °C for 1 min, 50 °C for 30 s, and 72 °C for 1 min, with a final extension at 72 °C for 7 min. For FCoV-Mut-F7/R7 and FCoV-Mut-F8/R8 primers, the nested PCR reaction was performed at 95 °C for 5 min, followed by 35 cycles at 94 °C for 1 min, 55 °C for 1 min, and 72 °C for 10 min. PCR products were then purified using the QIAquick PCR Purification Kit (QIAGEN, USA). After purification, PCR products were sent to 1st BASE company, Malaysia, for sequencing.

**Table 1 viruses-17-00188-t001:** Primers used for PCR and nested PCR in this study.

Name of Primers	Nucleotide Sequence (5′–3′)	PCR Product (bp)	References
FCoV-P205	GGC AAC CCG ATG TTT AAA ACT GG	223	[16]
FCoV-P211	CAC TAG ATC CAG ACG TTA GCT C		
Iffs	GTT TCA ACC TAG AAA GCC TCA GAT	Type I: 376 Type II: 283	[15]
Icfs	GCC TAG TAT TAT ACC TGA CTA		
Iubs	CCA CAC ATA CCA AGG CC		
nIffles	CCT AGA AAG CCT CAG ATG AGT G	Type I: 360 Type II: 218	
nIcfs	CAG ACC AAA CTG GAC TGT AC		
nIubs	CCA AGG CCA TTT TAC ATA		
FCoV-Mut-F5	CAA TAT TAC AAT GGC ATA ATG G	598	[12]
FCoV-Mut-R5	CCC TCG AGT CCC GCA GAA ACC ATA CCT A		
FCoV-Mut-F6	GGC ATA ATG GTT TTA CCT GGT G	142	
FCoV-Mut-R6	TAA TTA AGC CTC GCC TGC ACT T		
FCoV-Mut-F7	GGC AGA GAT GGA TCT ATT TTT GTT A	1.582	
FCoV-Mut-R7	ATA ATC ATC ATC AAC AGT GCC		
FCoV-Mut-F8	GCA CAA GCA GCT GTG ATT A	156	
FCoV-Mut-R8	GTA ATA GAA TTG TGG CAT		

### 2.5. Data Analysis

Nucleotide sequences were aligned and analyzed using the BioEdit software version 7.2 [17] and CLUSTAL W Tool [18]. Nucleotide homology between the obtained sequences and those published in GenBank (Table 2) was carried out using GENETYX v.10 (GENETYX Corp., Tokyo, Japan) and the BLAST program (https://blast.ncbi.nlm.nih.gov/Blast.cgi, accessed on 15 July 2024). A phylogenetic tree was built using the Kimura 2-parameter model [19] and maximum likelihood estimation on the MEGA X software (https://www.megasoftware.net/, accessed on 15 July 2024) with a bootstrap value of 1000. The accession numbers of the Vietnamese FCoV sequences used in this study were PQ824910 to PQ824918.

### 2.6. Statistical Analysis

Fisher’s exact test was used to compare the rate of positive samples for the FCoV genome. A *p*-value less than 0.05 was considered statistically significant.

## 3. Results

### 3.1. Detection of FCoV Genome in Field Samples Using PCR

To assess FCoV infection in fecal samples collected in four provinces/cities in northern Vietnam, conventional PCR was used, and the results are indicated in Table 3. Of the 166 samples tested, 19 (11.45%) were positive for the FCoV genome using conventional PCR. According to the regions, the rates of gene-positive samples in Bacgiang and Hanoi were 17.14% and 12.5%, respectively. These rates were significantly higher than those in Hungyen (8.33%) and Hanam (5.26%) (Table 3). We also detected FCoV infection among cats in northern Vietnam according to breeding, age, gender, and health status. There were insignificant differences (*p* > 0.05) in the percentage of gene-positive rates between types of breed and age. FCoV genomes were detected in both healthy and sick cats (with diarrheal and non-diarrheal clinical signs) in northern Vietnam (Table 4).

### 3.2. FCoV Typing, Genetic, and Phylogenetic Analysis

Nested PCR was used to identify the typing of FCoV strains in this study, as previously described [15]. The results indicated that all 19 FCoV-positive samples were of type I FCoV.

Of the 19 FCoV-positive samples, 9 samples were randomly selected for nucleotide sequencing of the partial S gene sequence (120 bp). The nine Vietnamese FCoV strains were designated as Feline/Vietnam/FCoV/VNUA-10, -16, -19, -22, -34, -41, -51, -66, and -119. Among the Vietnamese FCoV strains obtained, nucleotide identity ranged from 88.33% (Feline/Vietnam/FCoV/VNUA-10 and -16 vs. Feline/Vietnam/FCoV/VNUA-51) to 99.16% (Feline/Vietnam/FCoV/VNUA-22 vs. Feline/Vietnam/FCoV/VNUA-34) (Table 5). The nine FCoV strains were also compared with other strains from abroad in GenBank. The highest nucleotide identities were 93.51% (Feline/Vietnam/FCoV/VNUA-51 vs. Feline/China/CD0617/2020), 93.75% (Feline/Vietnam/FCoV/VNUA-10 vs. Feline/Netherlands/FECV351/2012), 96.25% (Feline/Vietnam/FCoV/VNUA-19 vs. Feline/Netherlands/FECV875/2012), 96.80% (Feline/Vietnam/FCoV/VNUA-16 and Feline/Vietnam/FCoV/VNUA-34 vs. Feline/Netherlands/FIP321/2012), 97.11% (Feline/Vietnam/FCoV/VNUA-66 vs. Feline/China/LS0612/2020), 97.16% (Feline/Vietnam/FCoV/VNUA-119 vs. Feline/China/LS0612/2020), 97.50% (Feline/Vietnam/FCoV/VNUA-41 vs. Feline/Netherlands/FECV407/2012), and 97.87% (Feline/Vietnam/FCoV/VNUA-22 vs. Feline/Netherlands/FIP321/2012) (Table 6).

A phylogenetic tree was constructed based on the current nine FCoV strains and 55 sequences downloaded from GenBank, as previously used by [14]. The phylogenetic analysis indicated that the nine Vietnamese FCoV strains belonged to type I (Figure 1). They were divided into several clades. The identified FCoV strains were genetically related to viral strains reported in Europe (The Netherlands, France, and Germany) and Asia (China and Taiwan) and differed from the vaccine strain (Feline/USA/DF2 (DQ286389.1)).

### 3.3. Detection of Mutation Sites 23,531 and 23,537 and Key Restriction Site Detection of Furin Protein in the S1/S2 Region of FCoV

The partial S gene (120 bp) was used to analyze mutation sites 23,531 and 23,537. Results indicated that nucleotides at sites 23,531 and 23,537 of the nine Vietnamese FCoV strains were Adenine (A) and Thymine (T), respectively (Figure 2). Furin cleavage in the S1/S2 site of the nine Vietnamese FCoV strains indicated an R/G-R-S/A-R-R-S motif (Figure 3). These mutations may be associated with changes of virulence of FCoV strains [12,13,20,21].

## 4. Discussion

FCoV strains are circulating and affecting domestic cats worldwide. The virus may cause FIP and mild diarrheal clinical signs in cats [9]. Although there is a vaccine available in some countries it is considered to have only limited efficacy in some situations [22]. Therefore, several therapeutics were studied to treat cats suffering from FCoV, based on the (i) non-specific modulation of the immune system, (ii) use of immunosuppressive drugs, and (iii) use of inhibitors to reduce viral replication [23,24,25]. In addition, biosafety and biosecurity measures are potential strategies to manage this pathogen [22]. To our knowledge, this is the first study to report on FCoV infection as well as the genetic characterization of viral strains in domestic cats in northern Vietnam. Regarding the prevalence of FCoV infection globally, FCoV viruses have been detected in many countries. Positive rates vary among countries, ranging from low (6.6%) to high (95%) [2,26,27,28,29,30,31]. In the current study, we found that the FCoV-positive rate was 11.45% among Vietnamese domestic cats, which is higher than the 6.6% reported in South Korea [31], but lower than the 95% detected among FIP cats in Portugal [32] or 74.6% to 80.35% in China [30]. Differences in viral positive rates are possibly due to the characteristics of the population sampled and the sampling strategy. Our findings suggest that FCoV strains are circulating and impacting domestic cats in Vietnam.

Previous studies have found that age, breed, and gender are associated with FCoV infection or the development of FIP among domestic cats [29,33,34,35]. Addie et al. noted that 4–5-week-old cats are susceptible to FCoV infection when maternal antibodies disappear. However, cats at two weeks of age can be infected with FCoV strains [35]. Li et al. reported that cats older than six months may be more susceptible to infection than younger cats [29]. In the present study, no relationship was found between the viral positive rates and the age, breed, gender, or health status of cats, which differs from findings among FIP and non-FIP cats in Thailand in 2019 [36]. Further studies need to expand the sample size and clarify these points. It has been reported that FCoV infection is detected in both diarrheal, FIP, and healthy domestic cats [29,31,36,37,38]. This characteristic of FCoV infection results in the difficulty of diagnosing diseased cats. In the present study, FCoV infection was found in both diarrheal (20%) and healthy (19.57%) cats, with an insignificant difference between the two groups. Therefore, continued studies at the molecular level of the Vietnamese FCoV strains are needed to develop prevention and control strategies for this disease in cats.

Phylogenetic analysis of the nine Vietnamese FCoV strains indicated that they were divided into several subclusters and were genetically related to viral strains from the Netherlands, France, Germany, China, and Taiwan. These findings suggest that these strains may have multiple origins and high diversity. A primary reason is that cats imported into Vietnam from various countries may carry virus strains from abroad. A commercial vaccine, developed based on a type II virus [39], was approved for import and marketing in Vietnam (personal communication). The current nine FCoV strains in this study differed from the vaccine strain, suggesting that the identified viral strains were field strains.

Chang et al. pointed out that mutations at sites 23,531 (A → T) and 23,537 (T → G), leading to a substitution at residue 1058 (Methionine to Leucine) on the S protein of FCoV strains, may result in the change of virulence from FECV to FIPV [12]. However, a later study noted that this substitution is associated with viral transmission from the intestine rather than the development of FIP [13]. Among the nine Vietnamese FCoV strains, no mutations were detected at sites 23,531 and 23,537. All identified strains were detected from the feces of non-FIP cats. Further studies should be conducted to collect samples from FIP cats to gain insight into the mechanisms of mutation points in the S gene on the virulence of viral strains.

It was reported that FECV strains consist of an S1/S2 cleavage site with the R-R-S-R-R-S motif, while the FCoV strains detected in FIP cats show several substitutions at sites P5 (R → K/G), P3 (S → A), P2 (R → H/P/L), P1 (R → T/G/M), and P1′ (S → L) [20]. In this study, the nine FCoV strains showed an S1/S2 cleavage site of the S protein with the R/G-R-S/A-R-R-S motif, which is typical for FECV strains. Two substitutions were detected at sites P5 (R → G) and P3 (S → A). The amino acid at site P1′ contributes to the cleavage of S1/S2 and is highly conserved among FECV strains. This site was also conserved among the Vietnamese FCoV strains. Our findings indicated that the S1/S2 protein sequences of the Vietnamese FECV showed high conservation, which is associated with virus replication in intestinal cells in the host [20].

## 5. Conclusions

In this study, the FCoV infection rate detected in fecal samples was 11.45% across four cities/provinces in northern Vietnam during 2022 to 2023. Breed, age, and gender were not associated with FCoV infection. The viral genome was detected in diarrheal, non-diarrheal, and healthy domestic cats raised in northern Vietnam. Based on the nested PCR and phylogenetic analysis of the partial S gene, type I FCoVs were the predominant strains circulating among Vietnamese cats in this study. No mutations at sites 23,531 and 23,537 were observed in the S gene sequences of the nine identified strains. The furin cleavage site in the S protein of the nine Vietnamese FCoV strains was R/G-R-S/A-R-R-S, which is typical for FECV strains.

## Figures and Tables

**Figure 1 viruses-17-00188-f001:**
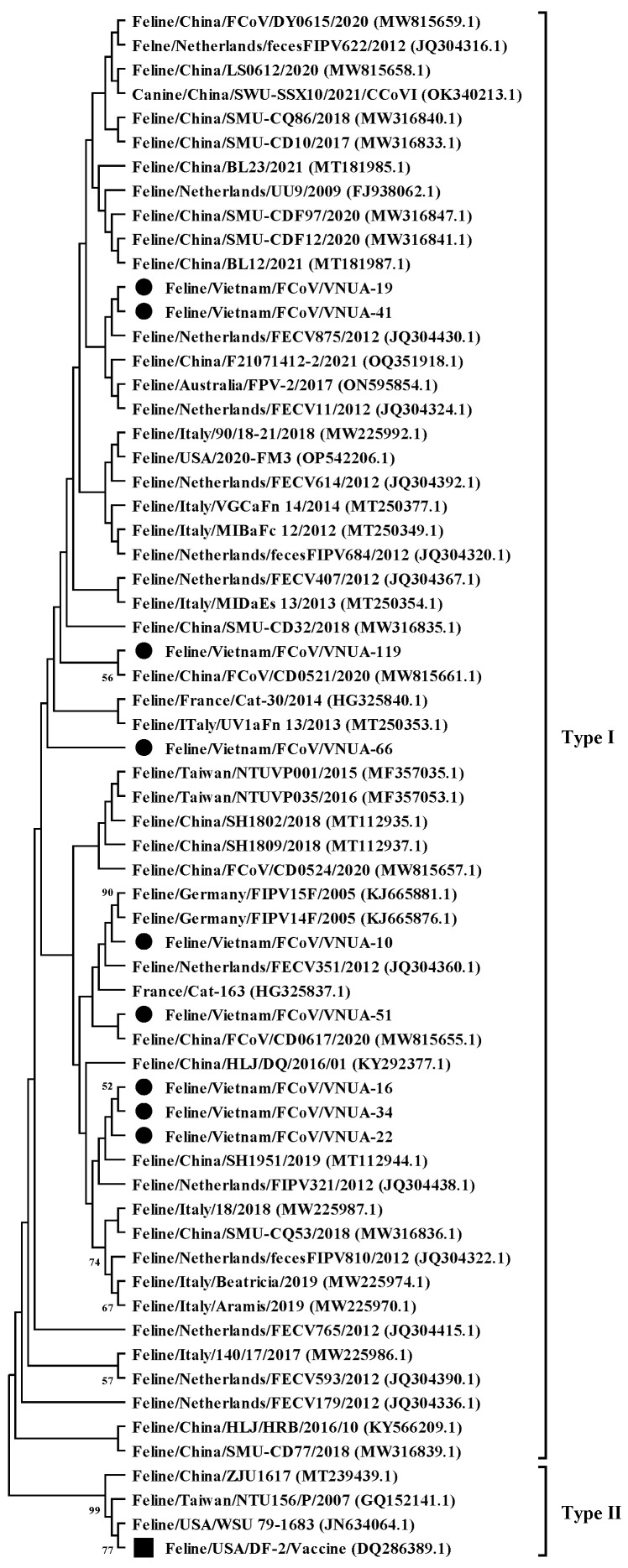
Maximum likelihood phylogenetic tree of partial S gene (120 bp) sequences of Vietnamese feline coronavirus strains compared with those available in GenBank. The MEGA X software maximum likelihood method was used to construct a phylogenetic tree (1000 bootstrap replicates). Numbers at each branch indicate bootstrap values of ≥50% by the bootstrap interior branch test. Vietnamese strains from this study are indicated by solid black circles, while the vaccine strain is indicated by a solid black square.

**Figure 2 viruses-17-00188-f002:**
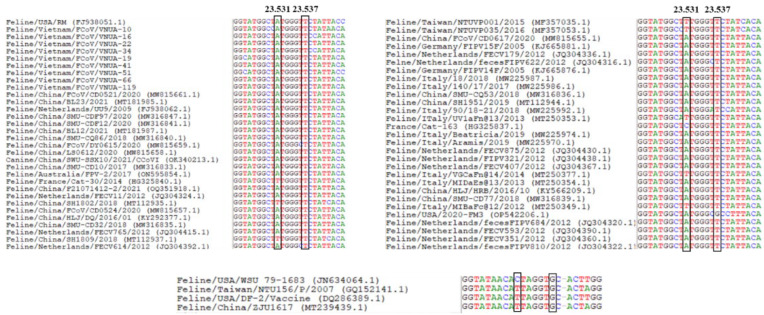
Detection of the mutation sites 23,531 and 23,537 of the Vietnamese FCoV strains.

**Figure 3 viruses-17-00188-f003:**
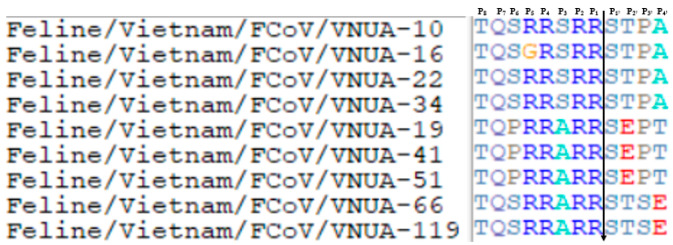
Detection of furin protein in the S1/S2 Region of the Vietnamese FCoV strains.

**Table 2 viruses-17-00188-t002:** Description of feline and canine coronavirus strains used in this study.

No.	GenBank Accession Number	Strain	Location	Host	Year	Type
1	MW316841.1	SMU-CDF12	China	Feline	2020	I
2	MT181987.1	BL12	China	Feline	2021	I
3	MW316847.1	SMU-CDF97	China	Feline	2020	I
4	MT181985.1	BL23	China	Feline	2021	I
5	MW316840.1	SMU-CQ86	China	Feline	2018	I
6	MW316833.1	SMU-CD10	China	Feline	2017	I
7	MW815659.1	DY0615	China	Feline	2020	I
8	MW815658.1	LS0612	China	Feline	2020	I
9	OK340213.1	SWU-SSX10	China	Canine	2021	I
10	MW316835.1	SMU-CD32	China	Feline	2018	I
11	MT112935.1	SH1802	China	Feline	2018	I
12	MT112937.1	SH1809	China	Feline	2018	I
13	KY292377.1	HLJ/DQ/2006/01	China	Feline	2016	I
14	MT112944.1	SH1951	China	Feline	2019	I
15	KY566209.1	HLJ/HRB/2016/10	China	Feline	2016	I
16	MW316839.1	SMU-CD77	China	Feline	2018	I
17	MW316836.1	SMU-CQ53	China	Feline	2018	I
18	OQ351918.1	F21071412-2	China	Feline	2017	I
19	MW815661.1	CD0521	China	Feline	2020	I
20	MW815657.1	CD0524	China	Feline	2020	I
21	ON595854.1	FPV-2	Australia	Feline	2017	I
22	MT250349.1	MIBaFc@12	Italy	Feline	2012	I
23	MT250353.1	UV1aFn@13	Italy	Feline	2013	I
24	MT250377.1	VGCaFn@14	Italy	Feline	2014	I
25	MW225992.1	90/18-21	Italy	Feline	2018	I
26	OP542206.1	FM3	USA	Feline	2020	I
27	HG325840.1	Cat-30	France	Feline	2013	I
28	KJ665881.1	FIPV15F	Germany	Feline	2005	I
29	KJ665876.1	FIPV14F	Germany	Feline	2005	I
30	HG325837.1	Cat-163	France	Feline	2013	I
31	MW815655.1	CD0617	China	Feline	2020	I
32	MF357035.1	NTUVP001	Taiwan	Feline	2015	I
33	MF357053.1	NTUVP035	Taiwan	Feline	2016	I
34	FJ938062.1	UU9	The Netherlands	Feline	2009	I
35	JQ304392.1	FECV614	The Netherlands	Feline	2012	I
36	JQ304320.1	fecesFIPV684	The Netherlands	Feline	2012	I
37	JQ304430.1	FECV875	The Netherlands	Feline	2012	I
38	JQ304316.1	fecesFIPV622	The Netherlands	Feline	2012	I
39	JQ304324.1	FECV11	The Netherlands	Feline	2012	I
40	JQ304360.1	FECV351	The Netherlands	Feline	2012	I
41	JQ304438.1	FIPV321	The Netherlands	Feline	2012	I
42	JQ304390.1	FECV593	The Netherlands	Feline	2012	I
43	JQ304336.1	FECV179	The Netherlands	Feline	2012	I
44	JQ304322.1	fecesFIPV810	The Netherlands	Feline	2012	I
45	JQ304415.1	FECV765	The Netherlands	Feline	2012	I
46	JQ304367.1	FECV407	The Netherlands	Feline	2012	I
47	MT250354.1	MIDaEs@13	Italy	Feline	2013	I
48	MW225986.1	140/17	Italy	Feline	2017	I
49	MW225987.1	18	Italy	Feline	2018	I
50	MW225970.1	Aramis	Italy	Feline	2019	I
51	MW225974.1	Beatricia	Italy	Feline	2019	I
52	DQ286389.1	DF-2	USA	Feline	2007	II
53	GQ152141.1	NTU156/P	Taiwan	Feline	2007	II
54	JN634064.1	WSU 79-1683	USA	Feline	2012	II
55	MT239439.1	ZJU1617	China	Feline	2016	II

**Table 3 viruses-17-00188-t003:** Detection of the feline coronavirus genome in field samples according to regions.

No.	Province/City	No. of Tested Sample	No. of Gene-Positive Samples (%)
1	Hanoi	64	8 (12.50) ^a,b^
2	Bacgiang	35	6 (17.14) ^a^
3	Hungyen	48	4 (8.33) ^a,b^
4	Hanam	19	1 (5.26) ^b^
Total		166	19 (11.45)

^a, b^ letters indicate significant differences between groups.

**Table 4 viruses-17-00188-t004:** Detection of the FCoV genome in field samples using PCR method according to breed, age, gender, and health status.

Criteria		No. of Tested Samples	No. of Gene-Positive Samples (%)
Breed	Native cats	36	6 (16.67)
	Exotic, cross-breed cats	130	13 (10.00)
Age (Months)	<6	24	2 (8.33)
	6–12	42	6 (14.29)
	>12	100	11 (11.00)
Gender	Male	93	13 (13.98)
	Female	73	6 (8.22)
Health status	Healthy	46	9 (19.57)
	Diarrheal clinical signs	15	3 (20.00)
	Non-diarrheal clinical signs	105	7 (6.67)

**Table 5 viruses-17-00188-t005:** Comparisons of nucleotide identity of partial S gene among sequences of nine Vietnamese FCoV strains obtained in this study.

Strain Name	Nucleotide Identity (%)								
	VNUA-10	VNUA-16	VNUA-22	VNUA-34	VNUA-19	VNUA-41	VNUA-51	VNUA-66	VNUA-119
VNUA-10	100								
VNUA-16	91.66	100							
VNUA-22	92.50	97.50	100						
VNUA-34	91.66	98.33	99.16	100					
VNUA-19	91.66	92.50	93.33	92.50	100				
VNUA-41	90.83	92.50	95.00	94.16	96.66	100			
VNUA-51	88.33	88.33	90.83	90.00	95.00	94.16	100		
VNUA-66	91.66	95.00	95.83	95.00	96.66	96.66	92.50	100	
VNUA-119	90.83	92.50	93.33	92.50	95.83	95.83	91.66	97.50	100

**Table 6 viruses-17-00188-t006:** Comparisons of nucleotide identity of partial S gene of sequences of nine Vietnamese FCoV strains with downloaded sequences from the GenBank database.

No.	Strain Name	Virus with the Highest Nucleotide Identity			
		Country	Strain Name	Year	%
1	Feline/Vietnam/FCoV/VNUA-10	The Netherlands	FECV351	2012	93.75
2	Feline/Vietnam/FCoV/VNUA-16	The Netherlands	FIP321	2012	96.80
3	Feline/Vietnam/FCoV/VNUA-22	The Netherlands	FIP321	2012	97.87
4	Feline/Vietnam/FCoV/VNUA-34	The Netherlands	FIP321	2012	96.80
5	Feline/Vietnam/FCoV/VNUA-19	The Netherlands	FECV875	2012	96.25
6	Feline/Vietnam/FCoV/VNUA-41	The Netherlands	FECV407	2012	97.50
7	Feline/Vietnam/FCoV/VNUA-51	China	CD0617	2020	93.51
8	Feline/Vietnam/FCoV/VNUA-66	China	LS0612	2020	97.11
9	Feline/Vietnam/FCoV/VNUA-119	China	CD0521	2020	97.16

## Data Availability

The data presented in this study are available within the article. Raw data supporting this study are available from the corresponding author.

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
