# Peer review of "Feline Coronavirus in Northern Vietnam: Genetic Detection and Characterization Reveal Predominance of Type I Viruses"

_viruses, 2025, doi:10.3390/v17020188_

Round 1
Reviewer 1 Report
Comments and Suggestions for Authors
This is a well written and informative study addressing feline coronaviruses in Vietnam. As described in the title/abstract the authors conducted genetic surveillance, finding 19 (11%) positive.
1) There seems to be an error in the abstract as the 19 positives are noted as type II and not type I viruses; for the final conclusion of the abstract it would be useful to note that sequences found were consistent with FECV/low-path viruses, and that it was healthy cats (not FIP cats) that were sampled (eg in the title)
2) It is not clear what is meant by “primitive” (line 56). Please rephrase – to ‘ancestral’ or similar
3) What is meant by ‘closely related’ ..to NL63 (line 72). Ref 11 seems to state “Pairwise similarity of the feline infectious peritonitis virus (FIPV) KU-2 strain S protein was unusually low (45% of identical residues) with homologs encoded by other group 1 coronaviruses”. Can the authors elaborate more on this statement, or is that that they a simply all alphacoronaviruses, and so more closely related than betacoronaviruses etc
4) Please specify that ‘current’ in the legend to Figure 1 means “strains from this study”
5) Is there a reason why Figure 2 uses nucleotide numbering and Figure 3 uses amino acid numbering, this is confusing; pleas also state more clearly for Fig 3 that the sequences found were consistent with FECV/low-path viruses, and include some citations
6) There is no mention of the widespread use of antiviral drugs to treat/manage FIP/FCoV (eg as an alternative to vaccines, in line 211)
7) Ref 10 in incomplete (please add Greene’s to the title; ie Pederson and Sykes in Greene’s Infectious Diseases of the Dog and Cat)
Author Response
We are grateful to the editor and reviewers for their valuable comments and helpful suggestions. We have revised the manuscript based on the reviewers’ comments. All changes in the revised manuscript are indicated by a red-font color. Below, we have provided our point-by-point responses to the editor’s and reviewers’ comments.
To Reviewer #1
This is a well written and informative study addressing feline coronaviruses in Vietnam. As described in the title/abstract the authors conducted genetic surveillance, finding 19 (11%) positive.
1) There seems to be an error in the abstract as the 19 positives are noted as type II and not type I viruses; for the final conclusion of the abstract it would be useful to note that sequences found were consistent with FECV/low-path viruses, and that it was healthy cats (not FIP cats) that were sampled (eg in the title)
We thank for the reviewer’s corection. The changes have been made in the revised manuscript.
2) It is not clear what is meant by “primitive” (line 56). Please rephrase – to ‘ancestral’ or similar
We thank for the reviewer’s suggestion. The change has been made in the manuscript.
3) What is meant by ‘closely related’ ..to NL63 (line 72). Ref 11 seems to state “Pairwise similarity of the feline infectious peritonitis virus (FIPV) KU-2 strain S protein was unusually low (45% of identical residues) with homologs encoded by other group 1 coronaviruses”. Can the authors elaborate more on this statement, or is that that they a simply all alphacoronaviruses, and so more closely related than betacoronaviruses etc.
We thank so much for the reivewer’s comment and suggestion. We deleted this sentence in the revised manuscript.
4) Please specify that ‘current’ in the legend to Figure 1 means “strains from this study”
We agree with the reviewer’s suggestion. The changes have been made in the revised manuscript.
5) Is there a reason why Figure 2 uses nucleotide numbering and Figure 3 uses amino acid numbering, this is confusing; pleas also state more clearly for Fig 3 that the sequences found were consistent with FECV/low-path viruses, and include some citations.
We thank for the reviewer’s comments on Figures 2 and 3. Sometimes, mutations at nucleotide is not resulted in amino acid substitution. Therefore, we performed at nucleotide level in Figure 2. Therefore, we would like to the reviewer to allow us keep this form in Figure 2.
We also provided more citations in the revised manuscript, following the reviewer’s suggestion.
6) There is no mention of the widespread use of antiviral drugs to treat/manage FIP/FCoV (eg as an alternative to vaccines, in line 211)
We thank for the reviewer’s suggestions. We modified the discussion to provide more information on treaing FIP cats in the revised manuscript.
7) Ref 10 in incomplete (please add Greene’s to the title; ie Pederson and Sykes in Greene’s Infectious Diseases of the Dog and Cat)
We thank for the reviewer’s comment and suggestion. We modified the reference in the revised manuscript.
Reviewer 2 Report
Comments and Suggestions for Authors
This paper investigates the prevalence and genetic characteristics of feline coronaviruses in North Vietnam. In this regard it is of some interest within the region re the prevalence of the virus in healthy and sick cats from four locations. The descriptive data is to some extent useful, but cannot be definitive in its conclusions because of the sample sizes and lack of examination for potentially confounding variables.
The genetic characterisation of the 19 of 166 samples positive by PCR is interesting and may be useful, particularly in relation to the region. Again, the data is essentially descriptive, comparing their findings to others; there are no new major findings, but still worth doing.
Feline coronavirus (FCoV) is quite a complicated virus in terms of its pathogencity. There are two types of the virus: type 1 is the predominant strain in cat populations, but type 11, a recombinant virus between FCoV and canine coronavirus, also occurs in cats though less commonly. In addition, both viruses can also be classified either as feline enteric coronavirus (FECV), or the much more virulent feline infectious peritonitis virus (FIPV) which is thought to occur through mutation of FECV allowing replication and spread in macrophages.
The authors are clearly aware of these distinctions, and their understanding of the literature and execution of the study is generally reasonable. However as detailed below, some points need to be clarified and there are some errors that need to be corrected.
The results show that all 19 positive samples were found to be type 1, and although the authors cite this eg in the abstract as serotype 1, it may be better just to state it is FCoV type 1. This also applies to line 52 where the word serotypes is used. This is because although there are papers showing serological differences between type 1 and 11, these are not clear cut in terms of pathogenesis and clinical situations.
Phylogenetically the 19 strains from Vietnam were similar to other type 1s from several other countries in parts of Europe and Asia. The authors also investigated 9 randomly selected samples for further genetic and phylogenetic analysis, comparing them to other virus sequences in GenBank. No mutations were detected in the two sites in the S gene sequence which have been suggested as being linked to the ability of the virus to spread into macrophages. In addition, the S1/S2 cleavage site motifs were typical of FECVs rather than FIPVs.
Minor points.
Line 21 in the simple summary. Should be ‘belonged to type 1’ ie not type 11. I guess this is just a typo.
Line 32. 85.5% should be 88.5%. Again a typo…..more proof reading advised!
Table 1. Bit confused re the primers with ref 17 – I may have missed this but is it referred to in the text?
Line 148. Refers to Table 1 – think this should be Table 3?
Line 152. Should this also be Table 3?
Line 155 – types of breed. ie not ‘groups of breeding’.
Line 156 – Table 4?
Lines 161- 163. Better to delete serotyping – see above. The types were identified by genotyping and phylogeny, not serology.
Line 211. Suggest eg ‘Although there is a vaccine available in some countries it is considered to have only limited efficacy in some situations’.
Line 221. Maybe replace with ‘…are possibly due to the characteristics of the population sampled and the sampling strategy’.
Line 227. …may be more susceptible to infection than…..
Line 238- 240. For some reason, the authors have got this wrong, although in the introduction (lines 61-63) they got it right. Essentially although type 1 viruses are more common than type 11 in cats, they can both be found in healthy, diarrhoeal, or FIP cases. The references here also need to be checked for accuracy and to provide some more appropriate ones. The comment on line 244- 246 about not being able to detect type 11 viruses because they had no samples from cats with FIP shows a slightly worrying lack of understanding.
Line 259 change ‘ in the intestine’ to ‘from the intestine’.
Author Response
We are grateful to the editor and reviewers for their valuable comments and helpful suggestions. We have revised the manuscript based on the reviewers’ comments. All changes in the revised manuscript are indicated by a red-font color. Below, we have provided our point-by-point responses to the editor’s and reviewers’ comments.
To Reviewer #2
Minor points.
Line 21 in the simple summary. Should be ‘belonged to type 1’ ie not type 11. I guess this is just a typo.
We thank for the reviewer’ comment and suggestion. The changes have been made in the revised manuscript.
Line 32. 85.5% should be 88.5%. Again a typo…..more proof reading advised!
We apologize for our mistake. The change has been made following the reviewer’s suggestion.
Table 1. Bit confused re the primers with ref 17 – I may have missed this but is it referred to in the text?
We thank for the reviewer’s comment. We provided the missing information on nested PCR for typing FCoV strains in the revised manuscript.
Line 148. Refers to Table 1 – think this should be Table 3?
We thank for the reviewer’s suggestion. The change has been made in the revised manuscript.
Line 152. Should this also be Table 3?
We thank for the reviewer’s correction. The change has been made in the revised manuscript.
Line 155 – types of breed. ie not ‘groups of breeding’.
We agree with the reviewer’s comment and suggestion. The change has been made in the revised manuscript.
Line 156 – Table 4?
We thank for the reviewer’s correction. The change has been made in the revised manuscript.
Lines 161- 163. Better to delete serotyping – see above. The types were identified by genotyping and phylogeny, not serology.
We agree with the reviewer’s comments and suggestions. The changes have been made in the manuscript.
Line 211. Suggest eg ‘Although there is a vaccine available in some countries it is considered to have only limited efficacy in some situations’.
We thank for the reviewer’s suggestion. We changed the sentence in the revised manuscript.
Line 221. Maybe replace with ‘…are possibly due to the characteristics of the population sampled and the sampling strategy’.
We agree with the reviewer’s suggestion. The changes have been made in the revised manuscript.
Line 227. …may be more susceptible to infection than…..
We thank for the reviewer’s suggestion. The change has been made in the manuscript.
Line 238- 240. For some reason, the authors have got this wrong, although in the introduction (lines 61-63) they got it right. Essentially although type 1 viruses are more common than type 11 in cats, they can both be found in healthy, diarrhoeal, or FIP cases. The references here also need to be checked for accuracy and to provide some more appropriate ones. The comment on line 244- 246 about not being able to detect type 11 viruses because they had no samples from cats with FIP shows a slightly worrying lack of understanding.
We apologize for this mis-understanding. We deleted this paragraph in the revised manuscript.
Line 259 change ‘ in the intestine’ to ‘from the intestine’.
We thank for the reviewer’s suggestion. The change has been made in the revised manuscript.
Additional change:
Some minor changes made were also highlighted in red color.